# Carbon Isotopes of Riparian Forests Trees in the Savannas of the Volta Sub-Basin of Ghana Reveal Contrasting Responses to Climatic and Environmental Variations

**Emmanuel Amoah Boakye [1,2], Aster Gebrekirstos [3], Dibi N'da Hyppolite [1], Victor Rex Barnes [2], Stefan Porembski [4] and Achim Bräuning [5,\*]**

[1]  West African Science Service Centre on Climate Change and Adapted Land Use Graduate Research Program in Climate Change and Biodiversity, Université Felix Houphouët Boigny, 01 BP V34 Abidjan, Cote D'Ivoire; eaaboakye@yahoo.com (E.A.B.); n_dibihyppolite@yahoo.fr (D.N.H.)
[2]  Faculty of Renewable Natural Resources, Kwame Nkrumah University of Science & Technology, 00000 Kumasi, Ghana; rexbee2000@yahoo.co.uk
[3]  World Agroforestry Center, 30677 Nairobi, Kenya; A.Gebrekirstos@CGIAR.ORG
[4]  Department of Botany, Institute of Biosciences, University of Rostock, 18051 Rostock, Germany; stefan.porembski@uni-rostock.de
[5]  Institute of Geography, Friedrich-Alexander-University Erlangen-Nürnberg, 91058 Erlangen, Germany
[\*]  Correspondence: achim.braeuning@fau.de

**Abstract:** Stable isotopes of tree rings are frequently used as proxies in climate change studies. However, species-specific relationships between climate and tree-ring stable isotopes have not yet been studied in riparian forests in the savannas of West Africa. Four cross-dated discs, each of *Afzelia africana* Sm. (evergreen) and *Anogeissus leiocarpus* (DC.) Guill. & Perr. (deciduous) in the humid (HSZ) and dry (DSZ) savanna zones of the Volta basin in Ghana were selected from a larger tree-ring dataset to assess the relationships between the tree-ring carbon isotope composition ($\delta^{13}$C values) and climatic parameters. The atmospherically corrected $\delta^{13}$C values of both studied species showed that *A. africana* was enriched in $^{13}$C compared to *A. leiocarpus*. Strong correlations were found between $\delta^{13}$C values of *A. africana* and *A. leiocarpus* with temperature, but weak correlations with precipitation. Spatial correlation analysis revealed significant relationships between $\delta^{13}$C values of both tree species and Sea Surface Temperatures in the Gulf of Guinea in the southern Atlantic Ocean. The results suggest that the carbon isotope composition of riparian trees in the Volta river basin has a potential to reconstruct climate variability and to assess tree ecological responses to climate change.

**Keywords:** riparian forests; savanna; dendrochronology; carbon isotopes; climate change; West Africa

## 1. Introduction

Riparian forests in arid and semi-arid regions are strips of woody vegetation growing along waterways [1,2]. Because they form linkages between terrestrial and aquatic ecosystems, riparian forests play an important role as ecological corridors and provide a variety of ecosystem services. They serve as important habitats, moderate stream temperatures for aquatic life and act as "ecological shelter" against desertification in the adjacent drylands [1,3]. Riparian forests trap seeds and filter sediment and nutrients transported from adjacent land areas into waterbodies [3]. This enables riparian areas to support higher plant productivity and biomass growth compared to non-riparian areas in the surrounding drier landscape [2,4]. Due to these and additional ecosystem functions, many riparian forests are protected by the Ramsar Convention and other legal acts by national regulations [3,4].

Riparian forests are controlled by hydrological processes [3] and as water is the most limiting resource to biological activity in drylands, the timing, and magnitude of ecosystem production, evapotranspiration and nutrient cycling are closely linked to precipitation [1,2,5]. Changes in hydroclimate will affect the size, frequency and seasonal timing of precipitation and moisture inputs into rivers. It is therefore anticipated that climate change will have a huge impact on vegetation distribution, growth and ecosystem functions of riparian forests in drylands [5,6]. However, despite their ecological relevance and climate change threats on riparian forests, this unique ecosystem is under-studied in the savannas of West Africa [1].

Just as trees in temperate regions, tropical tree species can also be interpreted as historical records of climatic signals because numerous species show common patterns in ring width variations that can be cross-dated [7–12]. Many studies have already successfully applied tree ring stable isotope analyses in tropical regions, most of them working on an annual resolution [6,13–15], and some studies reported stronger climatic signals in stable isotope chronologies compared to tree ring-width variations [6]. A strong negative correlation between annual precipitation and tree-ring $\delta^{13}$C values was found for several broadleaved tree species under various tropical climate regimes [6,13,14,16]. Gebrekirstos et al. [16,17] found significant negative correlations of annual $\delta^{13}$C values with humidity and positive correlations with temperature in the West African Sahel woodland. Because of such correlations, carbon isotope composition in tree rings is used for reconstruction of past climatic variations extending back into the pre-instrumental era and is therefore of special relevance in areas where instrumental climate records are short. Carbon isotope composition is also useful for evaluating the relative importance of natural variability and anthropogenic impacts on the global climate [18], and for predicting trees ecological responses to future climate conditions [19–23] by characterizing species water use efficiency and strategy [13,17].

The carbon isotope composition in tree rings results from $CO_2$ fractionation during photosynthesis at leaf level [24,25]. Carbon isotope fractionation is mainly due to (i) diffusion effect, when external $CO_2$ is transported through the boundary layer and the stomata into the internal gas space to carboxylation sites into the chloroplast, and (ii) at carboxylation sites because of the enzyme ribulose-1,5-bisphosphate carboxylase which discriminates against $^{13}CO_2$ due to the lower intrinsic reactivity of $^{13}$C [26,27].

Leaf habit of trees influences carbon isotope fractionation. Evergreen trees partly show higher $\delta^{13}$C values compared to deciduous species because they maintain their leaves during drier conditions, and stomata closure under dry conditions reduces the discrimination of $^{13}CO_2$ [6,28,29]. In the present study, we examined the tree rings of two riparian tree species, *Afzelia africana* (Fabaceae, evergreen) and *Anogeissus leiocarpus* (Combretaceae, deciduous) growing in the savannas of the Volta River basin of Ghana to document climatic variations. The species were selected because of their contrasting ecological attributes as well as their contribution to the livelihood and food resources of forest-dependent communities. Both species are widely distributed along riparian forests, have clearly detectable growth ring boundaries and are under threat from over-exploitation [11,30]. The Volta river basin is divided into three main agro-ecological zones which are aligned along a gradient of increasing moisture conditions from north to south: the Sudano-Sahelian, the Sudanian, and the Guinean savannas [31–33].

To assess the response of tree vegetation in the Volta river basin to climate and environmental changes, we studied two riparian forests in two ecological zones, one in the Guinean zone and one in the Sudanian zone. Those riparian forest ecosystems are embedded in the humid savanna zone (HSZ) and dry savanna zone (DSZ), respectively. The specific objectives of this study are (1) to determine if annual $\delta^{13}$C values of tree rings of *A. africana* and *A. leiocarpus* are related to functional differences in leaf phenology (evergreen or deciduous); (2) to investigate the coherence of $^{13}$C tree-ring signals between different sites (HSZ and DSZ); and (3) to assess whether similar climatic signals are recorded within the tree-ring $^{13}$C chronologies of the two species.

## 2. Materials and Methods

### 2.1. Study Area

The study area was located in the Volta basin of Ghana. The studied sites were situated in protected riparian forest reserves along the Afram and Tankwidi Rivers which flow through the humid savanna zone, HSZ (Guinean) and dry savanna zone, DSZ (Sudanian), respectively (Figure 1). Water levels of the two rivers are significantly reduced during the dry season which lasts from November–April in the Tankwidi catchment and from November–February in the Afram catchment. The two rivers are both 8–12 m wide. The dominant soil type in the floodplains is Luvisol [34], which is characterized by a sub-surface accumulation of clay and organic matter, and is composed of high activity clays with high base saturation. The soil physical structure is characterized by high porosity, good drainage and aeration [34]. The dominant woody species in the Afram riparian forests belong to the family Fabaceae and Combretaceae, whereas in the Tankwidi riparian forests, dominant families are Fabaceae and Rubiaceae [35,36].

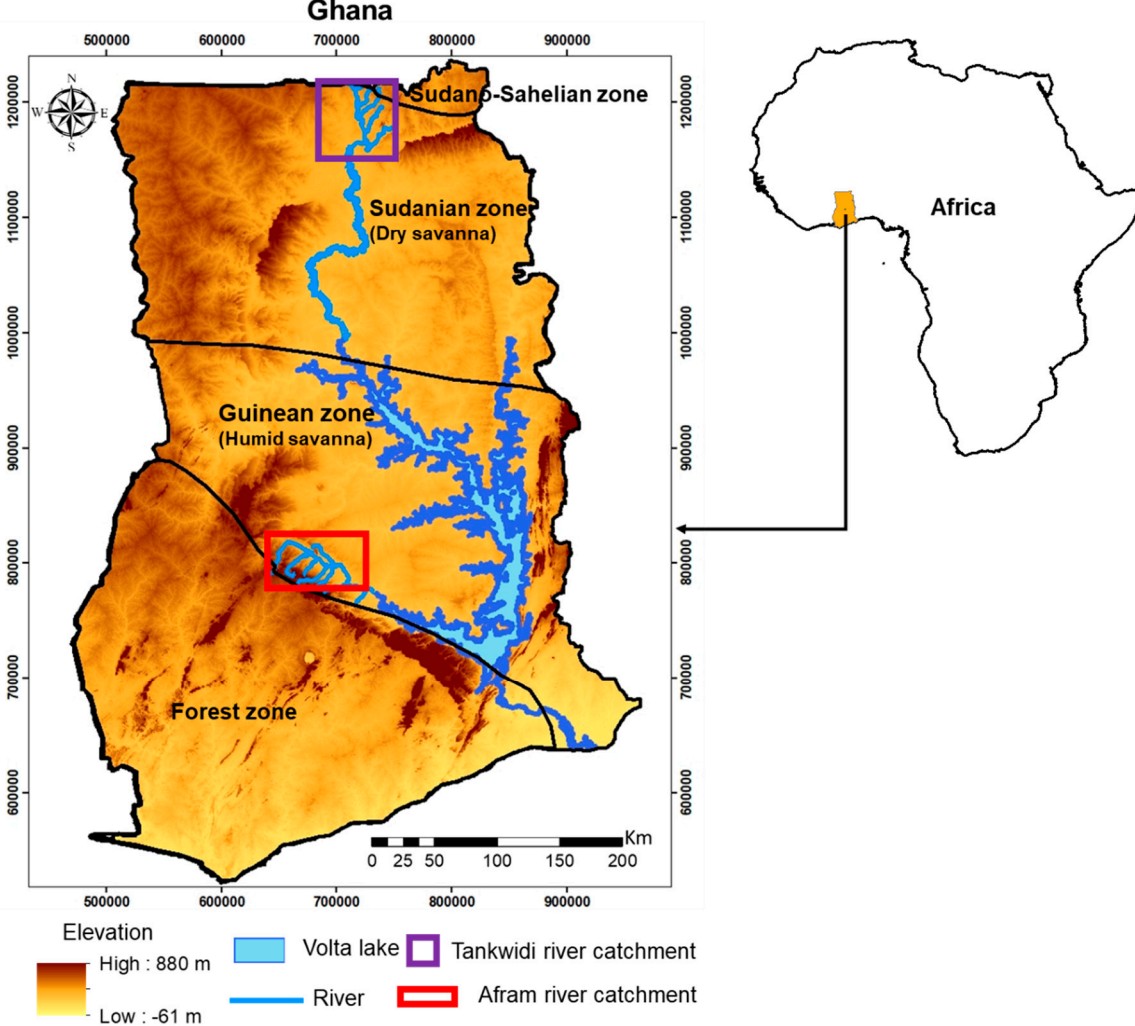

**Figure 1.** Location of the two study sites in two river catchments situated in different climatic zones of the Volta basin in Ghana, West Africa.

The seasonal movement of the inter-tropical convergence zone (ITCZ) and the related occurrence of the northeast trade winds determine the temporal and spatial distribution patterns of precipitation in the Volta Basin. Besides, atmospheric disturbances in the tropical Atlantic Ocean (TAO) determine

the occurrence of wetter and drier years in the study region. Sea surface temperatures (SSTs), both in the tropical Atlantic and the equatorial Pacific Oceans modulate frequency and strength of TAO disturbances [37]. The climate data covering the period 1961–2012 were obtained from the Navrongo (3 km to study site, Figure 2a) and Ejura (1 km to study site, Figure 2b) Meteorological Stations for the HSZ and DSZ, respectively. These were the climatic stations closer to the catchment areas as meteorological stations are sparse in Ghana. Mean annual maximum temperature in the HSZ reach 32 °C. Mean annual precipitation amounts 1100 mm. The, rainy season lasts from March to October, while precipitation rates occur in July and August. In the DSZ, mean annual maximum temperature reach 36 °C. Mean annual precipitation amounts 800 mm, and the rainy season lasts only seven months, from April to October. Correlation between precipitation and temperature data of both weather stations, Ejura ($r = -0.41$, $p = 0.33$) and Navrongo ($r = -0.13$, $p = 0.56$), showed weak negative correlations. Whereas a weak positive correlation was observed between mean annual precipitation data from Ejura ($r = 0.31$, $p = 0.03$) and Navrongo ($r = 0.33$, $p = 0.04$) weather stations with annual sea surface temperatures (SSTs) from 1961 to 2012. Similarly, mean annual maximum temperature showed highly significant correlations with SSTs for Ejura ($r = 0.59$, $p = 0.00$) and Navrongo weather stations ($r = 0.51$, $p = 0.00$) [11].

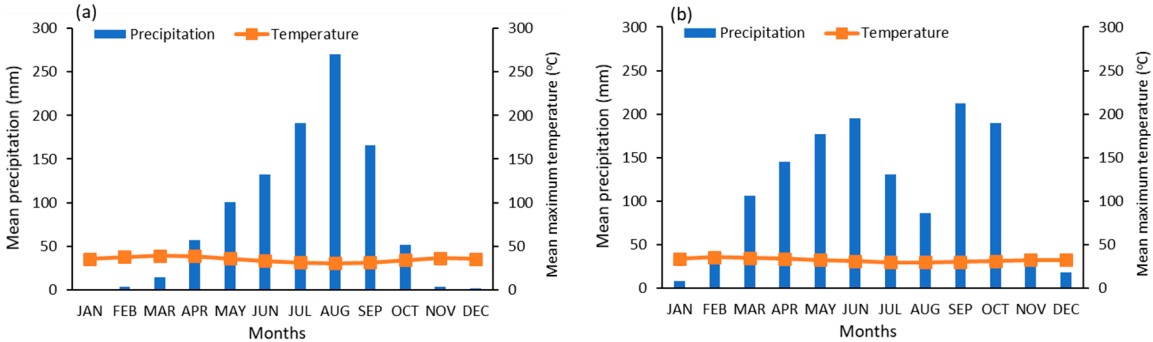

**Figure 2.** Seasonal patterns of mean monthly precipitation and maximum temperature (averages of 1961 to 2012) of the dry (DSZ) (**a**) and humid (HSZ) (**b**) savanna zones.

### 2.2. Tree-Ring Stable Isotope Analyses

We selected a total number of 16 discs (Table 1) from a larger tree-ring data set of 31 stem discs of *A. africana* and *A. leiocarpus* from both the DSZ and HSZ [11]. The discs were collected between January and February 2014, and the four oldest and well cross dated individual trees were selected for stable isotope analyses which were conducted at the tree-ring laboratory of the Friedrich-Alexander University of Erlangen-Nuremberg, Germany. Ring widths of all samples were measured on 2–4 radii and properly cross-dated before annual rings were separated with a scalpel for stable isotope analyses. Due to the diffuse porous wood anatomy of the studied species, the whole annual growth ring was used, resulting in an annual resolution of the final $^{13}$C chronologies. The last year (2013) of each series was excluded from analysis as ring width formation was not yet complete at the time of sampling [11].

Earlier studies have used bulk wood for the isotope analysis of tree rings. However, different wood components may differ in their isotopic composition [28,38]. This variation in the chemical composition of the wood may influence $\delta$-values. Cellulose extraction has disadvantages, such as the labour intensiveness and costly laboratory procedure, which sometimes limit the number of samples that can be processed in a research project [38]. It is worth mentioning that recent studies comparing the isotopic composition of different wood components (cellulose, lignin and bulk wood) have found a high correlation between cellulose and bulkwood $\delta^{13}$C values as well as between cellulose and lignin $\delta^{13}$C values [16,39–41]. In some cases, bulkwood $\delta^{13}$C values was found to be equally suitable and sometimes even showed higher correlations with climate parameters than cellulose $\delta^{13}$C values [42–44]. To test the effect of cellulose extraction on stable isotope variability, we separated the annually resolved wood samples for isotope measurements into samples of bulk wood and α-cellulose. Between 12 to

16 annual rings from the outer parts of each cross-dated disc of *A. africana* and *A. leiocarpus* in each savanna type (HSZ and DSZ) were used for this experiment. This allows testing if the offsets of $\delta^{13}$C values remained constant over time thus helping to decide the effectivity of $^{13}$C archives of whole wood for paleoclimatic reconstruction.

**Table 1.** Characteristics of carbon isotope composition of *Afzelia africana* and *Anogeissus leiocarpus* in the dry (DSZ) and humid (HSZ) savanna zones of Ghana.

| | Dry Savanna Zone | | Humid Savanna Zone | |
|---|---|---|---|---|
| | *A. africana* | *A. leiocarpus* | *A. africana* | *A. leiocarpus* |
| Whole wood ($\delta^{13}$C) | | | | |
| Number of rings measured | 101 | 41 | 91 | 76 |
| Mean (‰) | −25.54 | −26.29 | −25.32 | −26.54 |
| Std. Dev. (‰) | 0.31 | 0.34 | 0.22 | 0.41 |
| Maximum (‰) | −25.01 | −25.60 | −24.95 | −25.75 |
| Minimum (‰) | −26.09 | −26.71 | −25.72 | −27.19 |
| Correlation (ring width and $\delta^{13}$C) | −0.06 | 0.14 | −0.15 | −0.09 |
| *p*-value of correlation | 0.52 | 0.39 | 0.18 | 0.44 |
| Cellulose ($\delta^{13}$C) | | | | |
| Mean (‰) | −23.59 | −24.36 | −24.69 | −25.04 |
| Std. Dev. (‰) | 0.28 | 0.77 | 0.21 | 0.28 |
| Number of rings measured | 17 | 12 | 14 | 14 |
| Offset (Whole wood-Cellulose, ‰) | 1.95 | 1.93 | 0.63 | 1.50 |

For stable isotope measurements, powdered wood samples were produced along two radii of each disc using a micro drill with a diameter of 0.5 mm. The powdered samples were pooled into tin capsules and homogenized with a metal stick to represent the whole ring following the method described by Gebrekirstos et al. [6]. Subsamples from each year were weighed into tin capsules (0.4–0.5 mg). The cellulose extraction process was carried out according to the methods described in Wieloch et al. [45]. Accordingly, resin, fatty acids, etheric oils, and hemicellulose were extracted with a solution of 5% NaOH for 2 h at 60 °C. This operation was repeated twice. Then, lignin was extracted with 7% NaClO$_2$ solution for 40 h at 60 °C. Hemicelluloses were then extracted with 17% NaOH for 2 h at room temperature. A washing procedure was interposed between the different steps. Finally, samples were washed once with 1% HCl and three times with boiled de-ionized water (until pH 7 ± 1) and transferred from the filter funnels into Eppendorf tubes with 1 mL de-ionized water. Following Laumer et al. [46], ultrasonic homogenization was carried out for 15 s with a UP200s (Hielscher Ultrasonics GmbH, Berlin, Germany). After freeze-drying for 72 h in an ALPHA 1-4/2-4 LSC lyophilisation unit, the dried cellulose was then weighed to determine the yield and finally measured by isotope ratio mass spectrometer (Delta V Advantage, Thermo Electron, Bremen, Germany) coupled to a HekaTech Elemental Analyzer with a precision and accuracy of up to ±0.1% relative. The results are given in $\delta$-notation, which is the relative deviation from the PDB (Pee Dee Belemnite) standards:

$$\delta^{13}\text{C} = [(^{13}\text{C}/^{12}\text{C}) \text{ sample}/(^{13}\text{C}/^{12}\text{C}) \text{ PDB}) - 1] \times 1000‰$$

### 2.3. Statistical Analysis

Tree-ring isotope data may contain trends unrelated to climate but related to the decline in atmospheric $\delta^{13}$C values caused by burning of $^{13}$C-depleted fossil fuel. This trend was removed by following the procedure of McCarroll and Loader [28]. Analysis of variances was done to determine whether $\delta^{13}$C values differed across the different species; and the significant differences were tested through Tukey's pair-wise comparison. Pearson correlations were conducted between the $^{13}$C chronology of each species (*A. africana* and *A. leiocarpus*) of the HSZ and DSZ and total monthly and annual averages of precipitation and monthly maximum temperature. The average monthly minimum

temperature did not show correlations with $\delta^{13}$C values and are therefore not presented. The influence of regional SSTs on the $^{13}$C chronologies was evaluated by spatial correlation analysis using the KNMI Climate Explorer (http://climexp.knmi.nl/). Highly significant correlations of regional SSTs and the $^{13}$C chronology of each species were presented in the results.

## 3. Results

### 3.1. Relationship between $^{13}$C Chronologies of Whole Wood and Cellulose

Comparative analyses (Figure 3) of *A. africana* and *A. leiocarpus* from the HSZ and DSZ showed that the $\delta^{13}$C values of whole wood were lower than $\delta^{13}$C values of α-cellulose. $\delta^{13}$C values of whole wood and cellulose showed significant positive correlations for both species. The highest correlation was recorded for *A. africana* in the HSZ (r = 0.88, *p* = 0.00) followed by *A. africana* in the DSZ (r = 0.84, *p* = 0.00). *A. leiocarpus* of the HSZ (r = 0.59, *p* = 0.03) followed up third with *A. leiocarpus* in the DSZ (r = 0.40, *p* = 0.20) having the weakest correlations (Figure 3a,b). The mean offset of the $\delta^{13}$C values of whole wood and cellulose for *A. africana* in the HSZ was small (0.63‰), *A. africana* in the DSZ was 1.95‰, whereas *A. leiocarpus* in the DSZ and HSZ were 1.93‰ and 1.50‰ respectively.

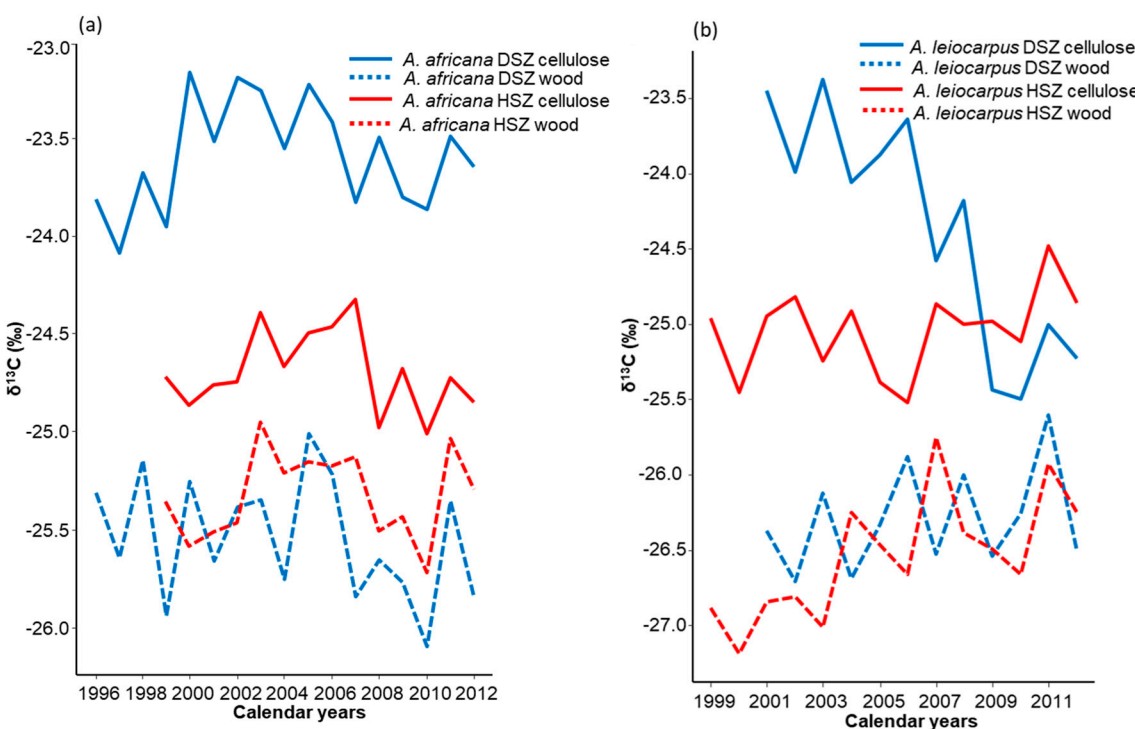

**Figure 3.** Pattern of annual $^{13}$C chronologies in whole wood and cellulose of *Afzelia africana* (**a**) and *Anogeissus leiocarpus* (**b**) in the dry (DSZ) and humid (HSZ) savanna zones of Ghana.

There was a significant difference in cellulose $\delta^{13}$C values among the species ($F_{(3, 53)}$ = 34.3, *p* = 0.00). Species-specific $\delta^{13}$C values (Table 1) showed that in the DSZ, the *A. africana* (evergreen) individuals are significantly (*p* < 0.05) enriched in $^{13}$C compared with *A. leiocarpus* (deciduous). This was, however, not the case for the HSZ despite *A. africana* having a higher $\delta^{13}$C values than *A. leiocarpus*. Comparison between the two savanna types also showed that the cellulose $\delta^{13}$C values of *A. africana* and *A. leiocarpus* of the DSZ are enriched in $^{13}$C (*p* < 0.05) compared to the same species in the HSZ. Correlation coefficients between the $\delta^{13}$C values of *A. africana* and *A. leiocarpus* collected from the HSZ (r = 0.45, *p* = 0.001) were significant, whereas they were weak in the DSZ (r = 0.10, *p* = 0.68). Significant correlations were also found between $\delta^{13}$C values of *A. africana* from the HSZ and the DSZ (r = 0.41, *p* = 0.001). In contrast, we found no significant correlation for *A. leiocarpus* between the HSZ

and DSZ (r = 0.13, *p* = 0.40). There was a weak correlation between species-specific ring widths and ¹³C chronologies for both HSZ and DSZ (Table 1).

### 3.2. Relationships between Tree-Ring $\delta^{13}C$ Values and Climatic Parameters

Despite site-specific differences, temperature and precipitation generally showed significant positive and negative correlations with $\delta^{13}C$ values of the studied trees (Figure 4). In the DSZ, *A. africana* showed positive correlations with temperature from January to July and September whereas *A. africana* in the HSZ showed positive correlations with temperatures throughout the year (Figure 4a). *A. africana* in the DSZ showed strong negative correlations with precipitation in July and October (Figure 4a). However, *A. africana* in the HSZ had a significant negative correlation with precipitation in January and March despite the low precipitation during those months (less than 30 mm of rains).

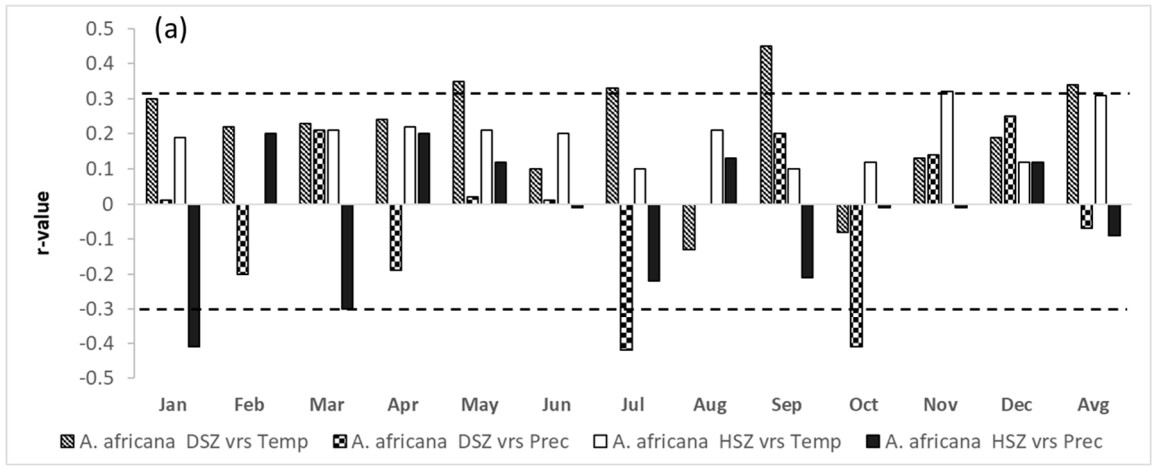

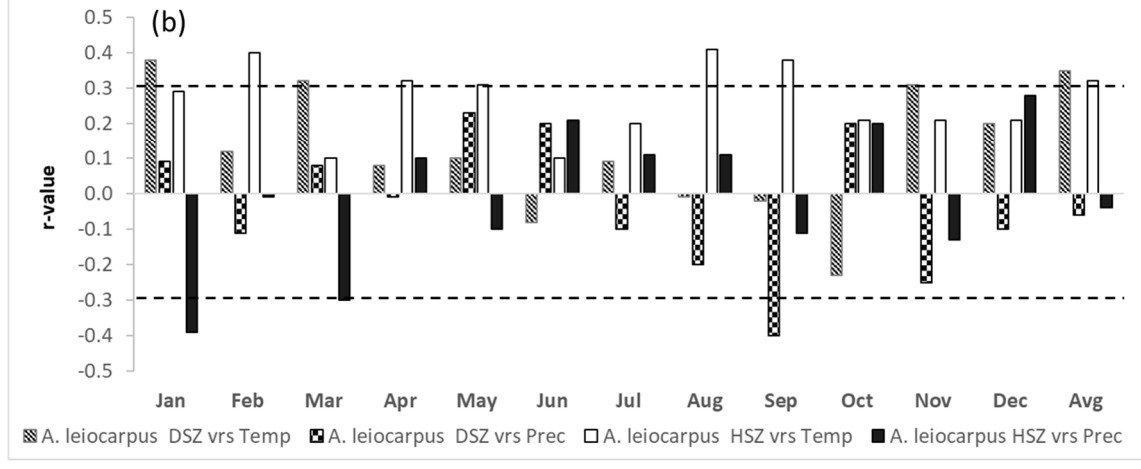

**Figure 4.** Correlations of *Afzelia africana* (**a**) and *Anogeissus leiocarpus* (**b**) chronologies from the dry (DSZ) and humid (HSZ) savanna zones with maximum temperature (temp) and precipitation (prec). For DSZ, wet period starts from April to October, and dry period from November to March. The wet period in the HSZ start from March to October, and dry period start from November to February. Avg signifies yearly averages. Dashed horizontal lines indicate significance ($p \leq 0.05$).

*A. leiocarpus* in the DSZ showed positive correlations with temperature from January to March and November. In October, the species showed negative relationships with temperature. *A. leiocarpus* in the HSZ however, showed a positive correlation with temperature throughout the year (Figure 4b). *A. leiocarpus* in the DSZ had a significant negative relationship with precipitation in September. In contrast, *A. leiocarpus* in HSZ had a significant negative correlation with precipitation in January and March.

Tree-ring $\delta^{13}$C values of *A. africana* and *A. leiocarpus* from the HSZ showed a positive correlation with gridded sea surface temperatures (SST) in the El Niño region of the equatorial Pacific Ocean and the Gulf of Guinea for the average values of the period, March to February (Figure 5). In the DSZ, both *A. africana* and *A. leiocarpus* showed a negative correlation with SSTs in the Gulf of Guinea and weak correlations with SSTs the Pacific region.

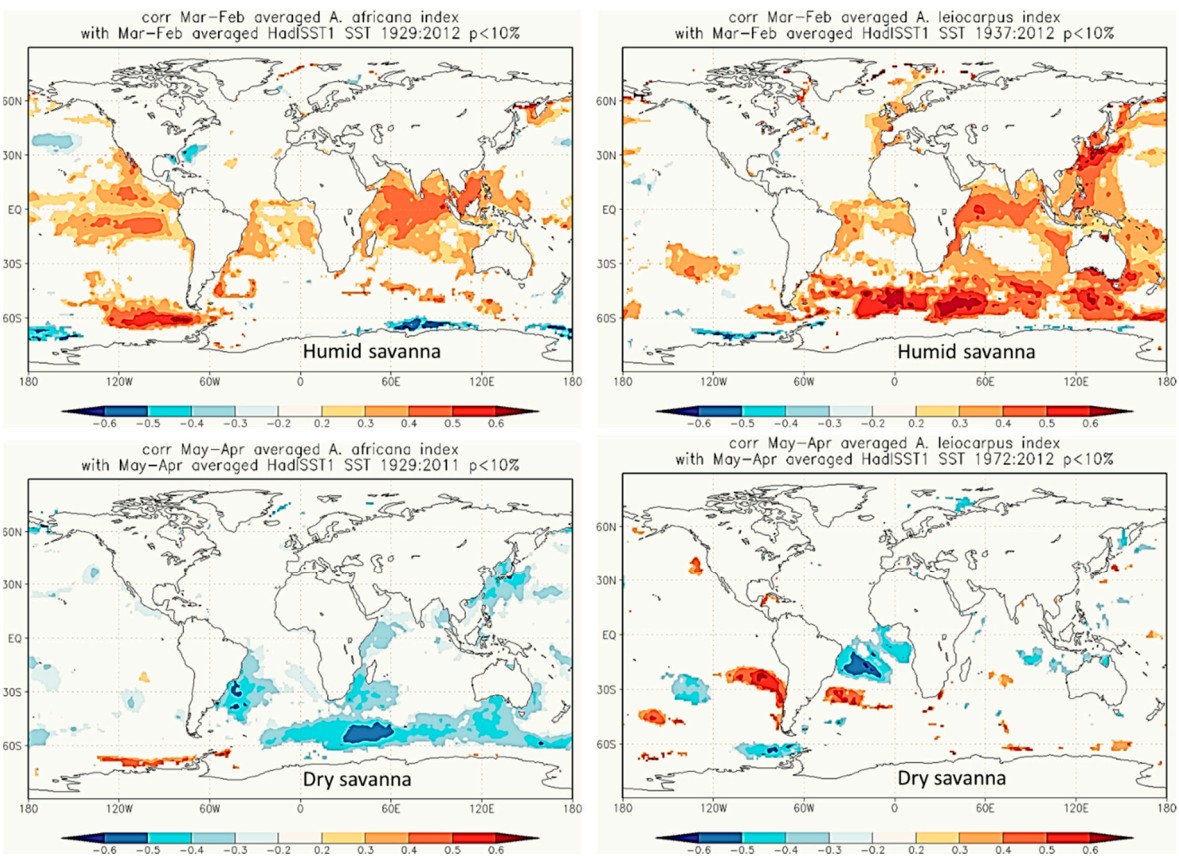

**Figure 5.** Spatial correlation of *Afzelia africana* and *Anogeissus leiocarpus* in both the humid (HSZ) and dry (DSZ) savanna zones with sea surface temperatures.

## 4. Discussion

### 4.1. Variation of $\delta^{13}$C Values of Bulk Wood and Cellulose in Tree Rings

Various wood components show different isotopic signals, and removal of wood constituents such as lignin and resins enhance the inter-annual variability of $\delta^{13}$C values in cellulose [43,47]. The means of the $\delta^{13}$C values of whole wood and cellulose of *A. africana* and *A. leiocarpus* in this study suggest (Table 1; Figure 3) that cellulose is enriched in $^{13}$C compared to whole wood, as already found in other studies [42,43,47–49]. Furthermore, the offset (~1–2‰) between whole wood and cellulose $\delta^{13}$C values of both species (Table 1) are in the range reported from the literature [40,42]. Because both study species (except *A. leiocarpus* in DSZ) showed constant differences in $\delta^{13}$C values of the whole wood and cellulose pattern, they are well suited as climate proxies [40]. Accordingly, whole wood was adopted for further exploration of the relationship of $\delta^{13}$C values of the species with climatic variables, thus simplifying the preparation procedure by skipping time-consuming cellulose extraction.

### 4.2. Patterns and Variations in Tree Rings $\delta^{13}$C Values in Different Tree Species

The higher enrichment of $^{13}$C in cellulose and whole wood of the evergreen *A. africana* in both the HSZ and DSZ zones indicates that the species may be more stressed at higher temperatures than

*A. leiocarpus* (Table 1; Figure 3). During long periods of high temperature, the deciduous *A. leiocarpus* avoids heat stress by shedding its leaves [29,50,51]. Because *A. africana* is evergreen, it can extend the length of the photosynthetic season. This presents challenges to the plants as they must tighten stomata closure to reduce water loss [6,51,52]. After the closure of the stomata, $^{13}CO_2$ which remains in the leaf air spaces and is less preferred by the RUBISCO enzyme is relatively less discriminated in the process of photosynthesis, thus increasing the proportion of $^{13}C$ in the leaf [53]. *A. africana* and *A. leiocarpus* in the DSZ (Table 1) had higher enrichment in $^{13}C$ compared to the same species in the HSZ probably because of enhanced water stress and stomatal limitations to photosynthesis. Stomata regulation is the strategy adopted by trees in drylands for minimizing evaporative water loss which could influence the enrichment of $^{13}C$ in dry savanna trees [21,51,54,55].

The low correlation between $\delta^{13}C$ values of *A. leiocarpus* and *A. africana* in the DSZ could have partly resulted from the differences in rooting depth or leaf architecture that cause the species to respond differently to climatic conditions. Evergreen species (*A. africana*) have the ability to reach the deeper layers of soil to access ground water to overcome dry seasons limitations than deciduous trees (*A. leiocarpus*) [51]. This was, however, not the case for the *A. leiocarpus* and *A. africana* in the HSZ, probably because of reduced spatial variability in nutrients and soil moisture of this riparian area [2].

### 4.3. Relationship Between Tree-Ring $\delta^{13}C$ and Climatic Parameters

The months during which the highest correlations occurred between tree-ring $\delta^{13}C$ values and climatic variables varied between the studied species. In most of the cases, temperature had a positive correlation with $\delta^{13}C$ values of *A. africana* (Figure 4a) and *A. leiocarpus* (Figure 4b). In dry years, leaf stomata may close and hence $^{13}C$ may be less discriminated. This increases the enrichment of $^{13}C$ in the tree rings and results in a reduction of tree growth [6,14].

The $\delta^{13}C$ values of the trees species showed weak sensitivity to precipitation, probably because of the influence of underground water or discharges from the upstream that controls the stream flow (Figure 4a,b). The correlation between precipitation and $\delta^{13}C$ values of *A. africana* in the HSZ and *A. leiocarpus* in the DSZ showed a better relationship to the rainy season (July–October), when trees are photosynthetically active and growth rates are high. The lack of general similarities in the response of the different species may be due to their differences in leaf phenology of the trees [14].

High SSTs in the Gulf of Guinea (Figure 5) are associated with high precipitation over large parts of West Africa, enhancing vegetation productivity [37,56,57]. In the DSZ, $\delta^{13}C$ values of *A. leiocarpus* and *A. africana* showed a negative correlation with the SSTs at the Gulf of Guinea probably because of the contribution to precipitation in the region and the impact in recharging the rivers to support the growth of trees in such dry landscape. *A. leiocarpus* and *A. africana* in the HSZ (which is much wetter than DSZ) showed a positive correlation with SSTs in the Gulf of Guinea, perhaps because excess precipitation can cause flooding in this kind of humid riparian forests. The long duration of flooding limits nutrient availability and gas exchange of plants which hinders the growth of trees [11,58]. Information on historic flooding episodes for the HSZ were not available for verification.

### 5. Conclusions

The present study is the first to assess the potential of tree-ring carbon isotope variations of riparian forest tree species for paleo-ecological research in Ghana. The $\delta^{13}C$ values of the riparian trees in the humid and dry savanna zones revealed differences in their responses to climatic and environmental fluctuations as the trees could be employing different strategies to overcome seasonal water limitations. *A. leiocarpus* strategy of leaf shedding and conserving water enables it to avoid temperature and water limitation stress better than *A. africana*. Increasing dry periods and decreasing precipitation in the dry savanna zone induce water stress among *A. africana* and *A. leiocarpus* growing in the riparian forests. Our study has also shown that $^{13}C$ signals of riparian trees in the humid and dry savanna zones of Ghana can be linked to climatic fluctuations. The analyses of the trees' carbon isotopes at an annual resolution should, however, be expanded in sample size and length for a robust

climatic reconstruction. For management purposes, carbon isotopes can be used for screening species for the restoration of degraded riparian landscapes in the savannas.

**Author Contributions:** E.A.B.: Design, sample collection, analysis, and write-up; A.G.: Design, analysis and write-up; D.N.H.: sample collection and analysis; V.R.B.: sample collection and analysis; S.P. and A.B.: Design, analysis and write-up.

**Funding:** This work was supported by the West African Science Service Center on Climate Change and Adapted Land Use (WASCAL) project of the German Federal Ministry for Education and Research.

**Acknowledgments:** The authors wish to express their sincere gratitude to the technical staff, Melanie Viehauser, Iris Burchardt, and Roswitha Höfner-Stich in the dendrolab of the Institute of Geography at Friedrich-Alexander University Erlangen-Nuremberg, Germany for their support during the laboratory work of the corresponding author in Germany.

**Conflicts of Interest:** The authors declare that they have no conflict of interest.

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
