# Peer review of "Carbon Isotopes of Riparian Forests Trees in the Savannas of the Volta Sub-Basin of Ghana Reveal Contrasting Responses to Climatic and Environmental Variations"

_forests, doi:10.3390/f10030251_

Round 1
Reviewer 1 Report
Dear Authors
Thank you very much for this manuscript, which is very interesting and has excellent potential and ecological relevance for the riparian forests of arid and semi-arid areas.
The subject of the manuscript is clear and well introduced.
Material and methods, and the area of study are clear and well described.
The objectives of the work are clearly described, but the discussion of the results and the conclusions are not always clear and well connected. I suggest carefully reviewing the discussion and conclusions and link it better to the showed results and the objective declared.
In particular, I suggest to review more closely the discussion about the pattern and variation of δ13C in the tree rings in the two tree species, and to highlight better the possible strategies adopted by the two species in two different environmental conditions. I think a more careful and deeper analysis of the isotopic data in relation to the main climatic parameters and leaf habit (leaf phenology), can help to better understand the actual adaptive capacity of the riparian tree species to climate/environmental changes and how the climate change can affect these riparian ecosystems.
The authors conclude that the two studied riparian species have different adaptive strategies because of the different leaf habit (but not only for this), but the adaptive potential in relation to climate changes in two different environments (dry and wet sites) is not well discussed.
Overall, it is a good manuscript of ecological importance, so I encourage the authors to make a further effort and to better implement the discussions and conclusions, keeping well in mind the objectives.
Details:
1 introduction
Line 73 “…Carbon isotope fractionation occurs during transpiration and photosynthesis at the…”
Carbon isotope fractionation occurs during the photosynthesis at leaf level (O'Leary, 1988; Brugnoli and Farquhar, 2000). Carbon isotope fractionation is mainly due to i) diffusion effect, when external CO2 is transported through the boundary layer and the stomata into the internal gas space to carboxylation sites into the chloroplast, and ii) at carboxylation sites because of the enzyme ribulose-1,5-bisphosphate carboxylase which discriminates against 13CO2 due to the lower intrinsic reactivity of 13CO2. While the water isotopes (18O and D) are fractionated at leaf level mainly due to the transpiration process (Ehleringer and Dawson, 1992; Farquhar and Lloyd, 1993). So please explain better how C isotope discrimination occurs.
Line 74 “..-leaf level and the leaf phenology has been found to influence the response of trees to changes in their environment…”
The concept of leaf phenology is complex, in the manuscript only one of the different aspects of phenology is taken into consideration, the leaf habit (evergreen and deciduous). I'm not sure it's right to talk about leaf phenology.
2. Materials and methods
2.1 Study area
146 -147 “Figure 2 Seasonal patterns of mean monthly precipitation and maximum temperature (averages of 1961 to 2012) of the dry (a) and humid (b) savanna zones.”
I suggest to modify (reduce) the y-axys scale for temperature to highlight the differences between the two sites. Moreover, it would be better to insert a color figure with a higher resolution, to simplify the reading of the figures.
I recommend to always use the same scale for the same parameters in the figures.
In the manuscript, the mean maximum monthly temperature is considered, why is it not considered the mean monthly temperature that generally affects more the daily photosynthesis rate and so the isotopic composition of the plant?
3 Results
3.1 Relationship between δ13C series of whole wood and cellulose
215- 2018 “The highest correlation was recorded for A. africana in the humid savanna (r=0.88, p=0.00) followed by A. africana in the dry savanna (r=0.84, p=0.00). A. leiocarpus of the humid savanna (r=0.59, p=0.03) followed up third with A. leiocarpus in the dry savanna zone (r=0.40, p=0.20) having the weakest correlations (Figure 3a and 3b)”
I suggest to the authors to insert a figure to show the correlations described and highlight the differences between sites and between species.
218-221 “………The mean offset of the δ13C of whole wood and cellulose for A. africana in the dry savanna is small and around 1‰, whereas A. africana in the humid savanna and that of A. leiocarpus of the two savanna types were higher (ca. 2 ‰)”
Data on figure 3 seem do not be coherent with values in table 1, please check.
In the figure 3b) it is interesting to note how the offset between δ13C of the whole wood and cellulose tends to decrease over time in both sites from a certain period onwards, in particular for the dry site, but this is not discussed.
243-244 " Table 1 Characteristics of δ13C series of Afzelia africana and Anogeissus leiocarpus in the dry and humid savanna zones of Ghana"
Table 1 is not easy to read. For people who are not expert of dendrochronology, it is not easy to understand all terms. Moreover, I suggest to check the offset values indicated in the table, they are not always coherent with those written in the discussion.
3.2 Relationships between tree-ring δ13C and climatic parameters
258 – 263 “A. leiocarpus in the DSZ showed positive correlations with temperature from January to March and November. In October, the species showed negative relationships with temperature. A. leiocarpus in the HSZ however, showed a positive correlation with temperature throughout the year (Figure 4b). A. leiocarpus in the DSZ only had a significant negative relationship with precipitation in September. In contrast, A. leiocarpus in HSZ had a significant negative correlation with precipitation in January and March.”
Even if there is a figure, maybe, in addition to the month I would indicate if it is a dry or wet period for the given site.
279 -280 “Figure 5 Spatial correlation of Afzelia africana and Anogeissus leiocarpus in both the humid and dry savanna zones with Sea Surface Temperatures
I find the quality of the figure 5 is very low.
4 Discussion
4.2 Patterns and variations in tree rings δ13C in different tree species
306- 308 “A. africana and A. leiocarpus in the dry savanna (Table 1) had low δ13C enrichment values probably because they may be tapping from ground water to reduce the temperature stress imposed on them.”
Actually for A. leiocarpus there are no significant variations between the dry and humid site.
306 – 317 “A. africana and A. leiocarpus in the dry savanna (Table 1) had low δ13C enrichment values probably because they may be tapping from ground water to reduce the temperature stress imposed on them. The trees could also be exhibiting the distinctive features of savanna trees such as smaller leaf sizes which result in a thinner boundary layer and allow greater vapor pressure for photosynthesis and greater sensible heat loss than enable them to cope with temperature stress [50]. The smaller leaf size also allows the species to adjust their leaf display more precisely under variable moisture conditions at low cost by rapidly dropping or reducing leaf area. Furthermore, dry savanna species are known to have higher leaf P and K concentrations that control the turgidity of stomatal guard cells and regulate gas exchange through the stomata. Increased nutrient concentrations may allow species to close stomata and reduce water loss rapidly under low moisture conditions. Maintaining cellular turgor through increased solute concentration can also help plants to cope with water stress”
Check this paragraph, the concepts are there, but it's a bit confusing. What are the possible differences between the two species between the two leaf habits in the same dry site?
4.3 Relationship between tree-ring δ 13C and climatic parameters
337… relationship to the rainy season (July-September), when trees are photosynthetically active…
Better to replace September with October.
Author Response
Dear editor and reviewers,
Thank you for your comments on our manuscript. We wish to submit a revised version of the manuscript for your consideration. We have improved the discussions and indicated the strategies used by the species to adapt to climatic and environmental changes. We also corrected the grammatical errors indicated in the text, figures and table. Changes in the manuscript has been highlighted in yellow. Details on other specific comments are indicated below. We look forward to the outcome of your assessment.
Yours sincerely,
Emmanuel Amoah Boakye
On behalf of co-authors
Review Report Form 1
Details:
1 introduction
Line 73 Suggestion of reviewer elaborating on carbon isotope fractionation was added to the manuscript..
Line 77 Leaf habits and carbon isotope fractionation has been addressed.
2. Materials and methods
2.1 Study area
Line 137 The scales of Figure 2 has been harmonized according to the suggestion of the reviewer.
3 Results
3.1 Relationship between δ13C series of whole wood and cellulose
Lines 204-220 The figure has been verified and align to the text
Lines 204-220 The text has been aligned to the δ13C values indicated in the table 1. Corrections were also made in the mean offset values as observed by the reviewer
3.2 Relationships between tree-ring δ13C and climatic parameters
Line 257 The figure 4 caption has been improved to include the wet and dry periods
Line 269 The resolution of figure 5 has been enhanced to make it more visible.
4 Discussion
4.2 Patterns and variations in tree rings δ13C in different tree species
Lines 286-306 was extensively revised taking into considerations the various suggestions of the reviewer
4.3 Relationship between tree-ring δ 13C and climatic parameters
Lines 316 The corrections on the months were effected.
Reviewer 2 Report
Dear Authors,
Thank you for the opportunity to read your manuscript. I found it interesting and believe this work makes a nice contribution to previous published works.
Key Message: I like what the message states but I don’t see it reflected clearly enough in the conclusions, which meet the three stated goals in the introduction text, but do not explicitly state potential influence of climate change on riparian forests. It does hint that both species are suitable candidates for restoring forests, due to their strategies. I think the Key Message either needs to be changed to better reflect existing conclusions, or (my preference) would be that the conclusions are expanded slightly to better reflect what the authors see as the potential way that stable isotopes can actually help us understand how climate change is influencing riparian forests. I think the content is already within the paper but it could be brought out more clearly.
Overall impression: There are some spacing issues in the references within the text. Some wording needs to be looked over, but the text reads well for the most part. It is an interesting study which will help lay the ground for others looking to better understand riparian trees and continue building our understanding of climate in West Africa, which is much needed.
The background provided a decent overview but could probably be improved with a bit more about the specific tree species used in the study.
Discussion and conclusions:
Overall the sections would benefit from clearer connection to the three stated aims.
Section 4.3: Lines 330 to 332 seem out of place within this paragraph. Reworking the text slightly would improve the flow. I think this section needs a clearer summation of what the tree carbon isotopes actually mean, and what the take away message is for those using data from tree isotopes in other contexts. I see that there is space for taking the conclusions in 347 and 348 and checking if there was flooding in any of the years the trees represent and seeing if there is a difference to other years where no flooding is suspected or reported. For me, this is a missing piece that needs to be added here which draws the line between the SST in Gulf of Guinea and actually what happened where the trees were growing, more than “probably”. If this isn’t possible, then tell the reader why it is not included. Revisit the monthly connections shown in Results 3.2 in this section more specifically.
Conclusions: By strengthening the message in the section 4.3, the conclusions will be more able to tie in to the key message.
Line 69: remove Only
Line 129: “Heavily sparse”. Remove Heavily.
205: “Peculiar” seems like an odd choice of words here. Would rework this sentence so its not in the negative (did not show, is not shown)
334: “such a drier” use “dry”.
Figure 1: Recommend an inset of location of Ghana on African continent. Much as I hope all readers would know where it is, this is an easy fix for those who don’t.
Figure 2: Recommend using same Y scale for both graphs, as this helps the reader quickly see the difference in monthly rainfall, with different scales they appear very similar at first glance. Place (a) and (b) on the left margin rather than centred above the graphs.
Figure 3: Recommend (a) and (b) moved to left margin.
Figure 4: Abbreviation of species not consistent with use in the rest of the manuscript.
Table 1: Expand explanation for those not working specifically with dendrochronology. Anyone should be able to read the table and know what “length” refers to in this context, but that is currently unclear.
Figure 5: Place correlation figures in the order in which they are mentioned in the text, eg reverse them to what they are currently.
Author Response
Dear editor and reviewers,
Thank you for your comments on our manuscript. We wish to submit a revised version of the manuscript for your consideration. We have improved the discussions and indicated the strategies used by the species to adapt to climatic and environmental changes. We also corrected the grammatical errors indicated in the text, figures and table. Changes in the manuscript has been highlighted in yellow. Details on other specific comments are indicated below. We look forward to the outcome of your assessment.
Yours sincerely,
Emmanuel Amoah Boakye
On behalf of co-authors
Reviewer form 2
Discussion and conclusions:
Line 321-330: Information on historic flooding was not available for the study sites for verification. This has been indicated in the text to make clear for readers.
Line 332-344 Conclusion was improved to include the strategies being used by the species to adapt to climate change. The role of carbon isotopes in screening species for ecological restoration of degraded riparian forests was also added.
Edits effected were as follows:
Line 69: remove Only
Line 129: “Heavily sparse”. Remove Heavily.
205: “Peculiar” seems like an odd choice of words here. Would rework this sentence so its not in the negative (did not show, is not shown)
334: “such a drier” use “dry”.
Figure 1: Recommend an inset of location of Ghana on African continent. Much as I hope all readers would know where it is, this is an easy fix for those who don’t.
Author: Figure 1 has been corrected
Figure 2: Recommend using same Y scale for both graphs, as this helps the reader quickly see the difference in monthly rainfall, with different scales they appear very similar at first glance. Place (a) and (b) on the left margin rather than centred above the graphs.
Author: Correction done
Figure 3: Recommend (a) and (b) moved to left margin.
Author: Correction done
Figure 4: Corrections done on Abbreviation of species to make consistent with the manuscript
Author: Correction done
Table 1: Elaborations made on terms in the manuscript
Author: Correction done
Figure 5: The Figure has been reversed according to the suggestion of the reviewer